# Not a ‘Straitjacket Affair’: Anthropometrically Derived Obesity Index Correlates of Elevated Blood Pressure among University Undergraduates

**DOI:** 10.3390/medsci5020009

**Published:** 2017-05-12

**Authors:** Chukwunonso E. C. C. Ejike, Patricia O. Ukegbu

**Affiliations:** 1Department of Medical Biochemistry, Federal University, Ndufu-Alike, Ikwo, PMB 1010 Abakaliki, Ebonyi State, Nigeria; 2Department of Biochemistry, Michael Okpara University of Agriculture, Umudike, PMB 7267 Umuahia, Abia State, Nigeria; 3Department of Human Nutrition and Dietetics, Michael Okpara University of Agriculture, Umudike, PMB 7267 Umuahia, Abia State, Nigeria; ukegbu.patricia@gmail.com

**Keywords:** anthropometry, blood pressure, correlations, hypertension, overweight, obesity

## Abstract

Obesity is known to correlate with measures of blood pressure (BP). The nature of the correlations has, however, remained a subject of scientific enquiry, especially when BP phenotypes are disaggregated and obesity is determined by a variety of methods. This study examined the relationship between obesity and BP in young-adult Nigerians. A total of 1610 subjects (53.9% females) were recruited from five universities in the Igbo-speaking part of Nigeria. Relevant BP and anthropometric data were obtained following standard protocols. Appropriate statistical tools were used for data analyses. The results show that 42.2% (49.5% males, 36.1% females) and 13.3% (15.2% males, 11.6% females) of the population had point prehypertension and hypertension, respectively. By body mass index (BMI) standards, 20.6% (12.4% males, 27.5% females) of the population were overweight/obese. Despite the weak positive and significant correlations between BP and the measures of obesity in both males and females in the general population (*r* = +0.110 to +0.261; *p* < 0.05), the correlations were found to exist essentially in normotensives, taper in the prehypertensives, and disappear (or became negative) among hypertensives. When analysed along weight status lines, a discordant relationship was found between the sexes. Overall, the relationship between blood pressure and measures of obesity is not linear throughout the BP spectrum. Clearly the said relationship is not a ‘straitjacket affair’.

## 1. Introduction

The relationship between obesity and elevations in blood pressure (BP) has recently been the subject of renewed scientific interest. Classical scientific thought proposes that weight gain ultimately leads to elevated blood pressures that, when sustained, results in hypertension [1]. This is founded on the understanding that adipose tissue hyperplasia/hypertrophy (especially around the viscera), which is central to obesity drives the mechanisms that lead to hypertension. For example, visceral adipose tissue-secreted adipokines are known to affect BP [2], while the altered renal sodium handling reported in obesity ultimately elevates BP in salt-sensitive individuals [3].

Conversely, there are reports of a ‘plateau’ or ‘hyperbole’ observed when the blood pressure versus body mass index (BMI) curve is split along blood pressure phenotypes [4,5,6]. These reports show that obesity (measured by BMI) is positively (and significantly) correlated with BP only among subjects with normal BP; and there is no significant correlation between BP and BMI among hypertensives. Implicit in the said reports is that the implications for weight loss may not be the same for subjects with normal BP and those with elevated BP.

Because the BMI is fraught with challenges, such as its inability to distinguish between muscle mass and adipose tissues, which makes its use in studying metabolic disorders difficult [7], it became necessary to investigate the relationship between BP and other measures of obesity that capture domains missed by BMI. This is important especially given the rising prevalence of hypertension even in low- and middle-income countries such as Nigeria. In fact, Ogah et al. [8] reported that ‘high blood pressure is the most common non-communicable disease in Nigeria’. An appropriate understanding of the relationship between hypertension and obesity, one of its most significant risk factors [9], will be useful in the prevention and management of the disease and in the allocation and mobilisation of scarce resources.

This study therefore tested the hypothesis that the plateau effect of high blood pressure on the blood pressure versus body mass index curves of adults exists independent of the measure of obesity in question. The results will hopefully broaden the understanding of the complex interrelationships between the two major contributors to cardiovascular disease—hypertension and obesity.

## 2. Materials and Methods

### 2.1. Study Design, Sample Size Determination, and Sampling Techniques

Five tertiary institutions in the five Southeastern States of Nigeria (one per State) were randomly selected for this cross-sectional study. The institutions are Ebonyi State University, Abakaliki, Imo State University, Owerri, University of Nigeria, Nsukka, Michael Okpara University of Agriculture, Umudike, and Anambra State University, Uli. Students attending these institutions of higher learning, aged 18 to 30, who gave informed verbal consent were recruited for the study. Pregnant or lactating women and subjects with any form of physical disability and/or overt or reported illness were excluded from the study. Ethical approval for this study was obtained from the Ethics Committee of the Federal Medical Centre, Umuahia on the 13th of June, 2014.

For each of the five institutions, 330 students who met the inclusion criteria were recruited. In each institution, subjects were purposively sampled to ensure that students from as many Colleges/Faculties and Departments were recruited. A total of 1610 subjects (53.9% females) out of the 1650 subjects recruited were eventually studied. Forty subjects had incomplete data entries and were excluded from the analyses.

### 2.2. Age Determinations and Anthropometric Measurements

Self-reported age at last birthday was recorded per subject. Heights of the subjects were measured with an inelastic vertical measuring rod to the nearest 0.1 cm. The subjects were required to stand on bare feet for all height measurements. For weight measurements, the subjects were required to be on bare feet and wearing light clothing, while a calibrated electronic bathroom scale (Starfrit 093826, Atlantic Promotions Inc., Quebec, QC Canada) was used. Weight measurements were made to the nearest 0.1 kg. The waist circumference (WC) of each subject was measured using a non-elastic fibre glass tape to the nearest 0.1 cm. The tape was snugly placed in a horizontal plane (parallel to the floor) round the abdomen, at the midpoint between the top of the iliac crest (just below the navel) and the lower margin of the last palpable rib in the mid axillary line. The hip circumference (HC) of each participant was measured at the widest circumference over the buttocks using a non-elastic measuring tape to the nearest 0.1 cm. From these measurements, waist-to-hip ratio (WHpR) was calculated as WC/HC; waist-to-height ratio (WHtR) was calculated as WC/Height; and BMI was calculated as weight (kg)/[height (m)^2^].

### 2.3. Blood Pressure Measurements

The blood pressures of the subjects were measured by trained personnel between 8 a.m. and 9 a.m. on study days. Subjects were required to be seated in a quiet room and rest for an initial 10 min. Thereafter, their blood pressures were measured using an oscillometric device (Omron HEM 7050, Omron Corp., Tokyo, Japan) with appropriate cuff sizes. Values for systolic blood pressure (SBP) and diastolic blood pressure (DBP) read off the device’s digital display were recorded for each subject. For each subject, three separate readings were taken, each at 5 min intervals. The first readings were discarded and the average of the last two readings recorded.

### 2.4. Definitions

Overweight/obesity was defined by multiple definitions, viz., (a) BMI ≥ 25 kg/m^2^ [10]; (b) WC > 102 cm for males, and WC > 88 cm for females [11]; (c) WHpR ≥ 1 [10]; (d) WHtR ≥ 0.5 [12]. Based on the blood pressure values, three blood pressure phenotypes were defined thus: normal (SBP/DBP < 120/80 mmHg); prehypertension (SBP/DBP ≥ 120/80 mmHg but < 140/90 mmHg); and hypertension (SBP/DBP ≥ 140/90 mmHg) [13].

### 2.5. Statistical Analyses

Data obtained from the respondents were analysed using IBM-SPSS version 20.0 (IBM Corp., Atlanta, GA, USA). Descriptive statistics were carried out and reported as mean ± standard deviation. Scatter plots of BMI vs. SBP/DBP curves were generated using Microsoft Excel (Microsoft Corp., Redmond, WA, USA). The one-way analysis of variance (ANOVA) test was used to determine if differences between continuous data for male and female respondents were significant. This was followed by post-hoc multiple comparisons using the least significant difference (LSD) method where necessary. Significant differences between categorical variables were tested for using the Fisher’s exact test (FET) or the Chi square (*X^2^*) test. The Pearson’s product moment correlation coefficients were calculated to assess the correlations between relevant variables. A *p*-value of less than 0.05 was accepted as statistically significant.

## 3. Results

As much as 42.2% of the population (49.5% for males and 36.1% for females; *X^2^* = 4.197, *p* = 0.040) had point prehypertension while 13.3% (15.2% for males and 11.6% for females; *X^2^* = 0.3857, *p* = 0.535) had point hypertension (Table 1). Systolic (pre)hypertension was more prevalent than diastolic (pre)hypertension. The mean SBP values (mmHg) for the males were 107 ± 8 (normotensives), 126 ± 7 (prehypertensives), and 140 ± 17 (hypertensives). For DBP in males, the values (mmHg) for normotensives, prehypertensives, and hypertensives were 70 ± 7, 79 ± 8, and 91 ± 10, respectively. For the females, the mean SBP values (mmHg) were 108 ± 9 (normotensives), 123 ± 9 (prehypertensives), and 139 ± 18 (hypertensives). For DBP in females, the values (mmHg) for normotensives, prehypertensives and hypertensives were68 ± 7, 79 ± 8, and 90 ± 16, respectively.

The prevalence of overweight/obesity expectedly varied depending on the index used to define it. WHpR diagnosed overweight/obesity the least (1.5%), while WHtR diagnosed it the most (22.7%). By BMI standards, 20.6% of the population (12.4% for males and 27.5% for females; *X^2^* = 7.793, *p* = 0.005) were overweight/obese, while by WC standards 8.6% of the population (0.8% for males and 15.3% for females; *X^2^* = 13.315, *p* < 0.001) were overweight/obese (Figure 1).

Hypertensive males had significantly higher heights, more weight, WC, HC, BMI, WHpR, and WHtR compared to males with normal blood pressure. In fact, they had significantly higher values compared to even the prehypertensive males on all the measured and determined parameters except age, height, and WHpR (Table 1). For the females, their ages and heights were statistically similar (*p* > 0.05), irrespective of the BP phenotype. BMI and WHpR were similar between female prehypertensive subjects and their counterparts with normal blood pressure. Female hypertensive subjects, however, consistently had significantly higher values compared to their prehypertensive counterparts (except for height) (Table 1). The reported anthropometric indices clearly increased significantly with the progression of BP phenotypes in the subjects, especially in the males.

Both SBP and DBP were positively and significantly correlated with all the measures of obesity in both males and females in the general population. The correlations were, however, all weak (*r* = + 0.110 to + 0.261). When the subjects were split along the BP phenotype lines, a different picture emerged. Significant positive correlations were found for subjects with normal BP, between SBP (but not DBP) and all the measures of obesity among the males. The same was true for the said measures except BMI in the females. Among prehypertensive subjects, SBP was correlated significantly and positively with only WC in both males and females. DBP was, however, correlated significantly and positively with all the measures of obesity in the females, and negatively with WHtR only in the males. Interestingly, among hypertensive subjects of both sexes, there was no significant correlation between SBP and any of the measures of obesity studied. DBP was significantly and negatively correlated with WC and BMI among the males, and positively with WHtR only in the females (Table 2). WHpR was found not to be correlated with any of the studied measures of obesity. Figure 2 shows the correlations between BMI and SBP/DBP in the studied population, irrespective of sex. From the figure, it can be seen that the positive correlation between BMI and BP disappears as one progresses along the BP spectrum and even becomes negative among hypertensive subjects.

Furthering the examination of the correlations between BP and measures of obesity, the subjects were stratified based on the weight status and the correlation coefficients calculated again. Given the low prevalence of overweight/obesity as determined by WC and WHpR, the calculations were restricted to only BMI and WHtR. The results show that, among normal weight subjects, SBP was correlated significantly only with WHtR and only in the females, while DBP was correlated significantly only with BMI and only in the males. Among the overweight/obese male subjects, SBP was correlated positively with only WHtR, while DBP was not correlated significantly with either measure of obesity. However, among their female counterparts, both BMI and WHtR were significantly and positively correlated with SBP and DBP (Table 3). Interestingly, none of the anthropometric predictors of obesity used in this study significantly predicted elevated SBP or DBP in either the males or the females in the general population (data not shown).

## 4. Discussion

Hypertension and cardiovascular disease are said to be related in an unbroken, persistent manner that is independent of other risk factors [8]. Owing to the high degree of morbidity and mortality attributable to hypertension and its sequelae, especially if untreated, and the need to identify (pre)hypertensive individuals early, it is important to quantify those who have the disease and then describe the relevant risk factors. As much as 42.2% of the studied population (49.5% for males and 36.1% for females) had point prehypertension, while 13.3% (15.2% for males and 11.6% for females) had point hypertension. A recent population-based study of men in Nigeria found hypertension and prehypertension to be present in 21.3% and 47.1% of the population, respectively [14]. These values are higher than the values reported here, especially with respect to hypertension. The age (which is a known risk factor for hypertension) of the subjects may be responsible for the disparity as this study recruited young adults, while the Ejike et al. [14] study recruited adults older than 40 years. Ukegbu et al. [15] had, however, reported a prevalence of 13.8% for hypertension in Nigerian adults, a figure similar to that reported here.

There are numerous other studies on hypertension in adult Nigerians. Two current systematic reviews on the subject will, however, suffice for comparative purposes. The first review reported a prevalence of 28.9% [95% confidence interval (CI): 25.1 to 32.8] [29.5% (95% CI: 24.8 to 34.3) for men and 25.0% (95% CI: 20.2 to 29.7) for women] [16]. The second review reported that the ‘crude prevalence of hypertension ranged from 6.2% (95% CI: 4.0 to 8.4) to 48.9% (95% CI: 42.3 to 55.5) for men and 10% (95% CI: 8.1 to 12) to 47.3% (95% CI: 43 to 51.6) for women’ [17]. The prevalence of hypertension reported in this study is lower than the values reported by the former authors, but falls within the lower values of the ranges reported by the latter authors. Obviously, despite the fact that more than a tenth of the studied population is hypertensive, the disease is still not as prevalent in this age bracket as it is in older subjects in Nigeria.

The large proportion of prehypertensive subjects reported here is however worrisome as they may eventually become hypertensive, especially with age. It is, however, plausible that the prevalence of both hypertension and prehypertension reported here may be related to ‘white coat’ hypertension as the subjects were largely naïve to blood pressure measurement. We found that there was a clear male preponderance of (pre)hypertension in the population. Akin to our findings, both systematic reviews [16,17] mentioned above also concluded that hypertension was more prevalent in men than in women. This is thought to be due to the fact that, prior to menopause, oestrogens in females are capable of modulating the renin-angiotensin-aldosterone system (RAAS) in a manner that reduces the development and/or progression of hypertension [18]. The finding of a preponderance of systolic hypertension is also a consistent finding in population-based BP studies [14,16,17].

Obesity is reported to be a significant risk factor for hypertension [9]. Though there are different anthropometric indices for the diagnosis of obesity, their individual effectiveness in identifying individuals that are predisposed to the obesity sequelae remain a subject of scientific debate. Expectedly, the different indices performed differently in diagnosing obesity/overweight. The BMI, however, remains the gold standard, despite its limitations [13]. This study found that WHtR diagnosed obesity/overweight the most (22.7%), while WHpR diagnosed the disease the least (1.49%). By BMI standards, however, 20.6% of the population (12.4% for males and 27.5% for females) were overweight/obese. We also found a clear female preponderance of overweight/obesity in the studied population. A recent study among young adults found that a total of 17% of the population (13% for males and 20.9% for females) were overweight/obese according to the BMI diagnostic criteria [19]. A slightly earlier study in young adults had reported a prevalence of 20.7% (17.5% for males; 24.8% for females) for overweight/obesity using the same standards [20]. The prevalence data reported here is therefore consistent with reports from their age-matched counterparts from Nigeria. Some studies in older Nigerian adults found an overweight/obesity prevalence of 15% and 42% in men and women, respectively [21] and 21.0% and 32.4% in men and women, respectively [15]. It therefore appears that the increase in obesity that comes with age may affect females disproportionately. Just as in this study, all four studies mentioned here reported a clear female preponderance of obesity in the populations they investigated.

The relationship between hypertension and obesity remains a subject of scientific controversy. While most authors agree that both disorders are correlated, the nature of such correlation is a matter of far less consensus [4]. In this study, we found that hypertensive subjects had significantly higher indices of obesity relative to those with normal BP. Furthermore, we found that, in the general population, blood pressures were positively correlated with measures of obesity. Ogah et al. [8] had reported modest correlations between blood pressure and measures of obesity in Nigerian adults. They however did not investigate the effect of disaggregating the data based on BP or weight phenotypes. When the data from the present study was however split along BP phenotypes, we found the ‘plateau effect’ reported in other populations in Nigeria [4,5,6]. Again, in a manner similar to the report of Ejike et al. [6], we found discordant correlations for males and females when the data is stratified along weight status lines. By the same token, the stronger relationship between blood pressure and measures of obesity in the females (especially when the data is stratified along weight status lines) is consistent with previous reports from Nigeria and elsewhere [6,22]. It therefore appears that there is a greater responsiveness to increasing adiposity in females than in males.

The implication of the observed ‘plateau effect’, especially among the males, is that the response of blood pressure to increases in obesity is not linear along the blood pressure or obesity continuum. The paradox of secular decreases in blood pressure, despite increasing obesity prevalence in economically developed societies, has been explained by suggestions that changes in dietary intake and/or exercise could influence BP through mechanisms not directly related to weight loss [23,24]. Bunker et al. [25] had suggested that there may be a BMI threshold at which its relationship with BP begins. There are also reports of excess cardiovascular mortality among lean hypertensive subjects [26] and in both thin and obese subjects [27,28]. Furthermore, Tesfaye et al. [29] after studying three different populations in Africa concluded that the risk of hypertension was not continuously distributed at all levels of BMI, despite a general increase in SBP and DBP with increasing BMI. These studies clearly suggest that the relationship between blood pressure and obesity is not a ‘straitjacket affair’. This is even more important as this study shows that the ‘plateau’ or ‘hyperbole’ exists for the relationship between BP and obesity, irrespective of the obesity-diagnostic index used. It therefore appears that a uniform weight loss requirement for the prevention and management of hypertension may not be proper for the entire population.

This study is limited by the use of an oscillometric device instead of the standard auscultation protocol. The device used here, however, has been validated in, and used extensively for, similar populations as the one studied. Secondly, blood pressures were measured on a single visit such that only point (pre)hypertension could be determined. The nature of this study and the limited resources at our disposal, however, were insurmountable constraints. Finally, the low levels of obesity and to some extent hypertension in the studied population may have made it difficult to appreciate the relationships studied here better. Subsequent studies should deliberately/purposively seek to recruit more overweight/obese and hypertensive subjects. The spread of our sampling and the depth of our analysis, however, are strengths of this study.

## 5. Conclusions

This study tested the hypothesis that the plateau effect of high blood pressure on the blood pressure versus body mass index curves of young adults exists independently of the measure of obesity in question. The results provide evidence for the hypothesis to be accepted. The relationship between blood pressure and measures of obesity is clearly not a ‘straitjacket affair’.

## Figures and Tables

**Figure 1 medsci-05-00009-f001:**
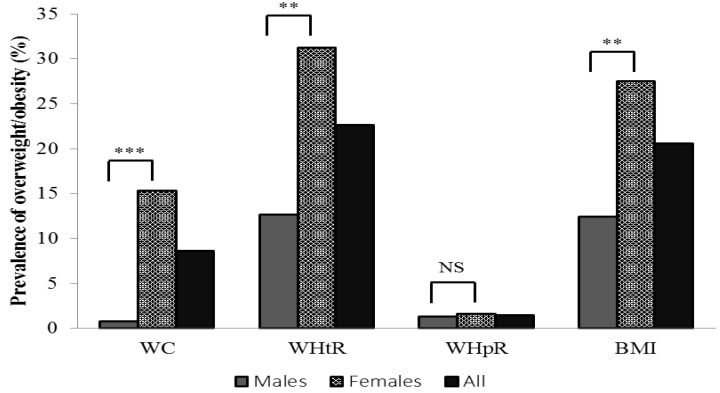
Prevalence of overweight/obesity determined by different anthropometric indices. NS, WC, WHtR, WHpR and BMI represent not significant, waist circumference, waist-to-height ratio, waist-to-hip ratio and body mass index respectively. *** and ** represent significant differences for Fisher’s exact test comparisons between males and females at *p* < 0.001 and *p* < 0.01, respectively.

**Figure 2 medsci-05-00009-f002:**
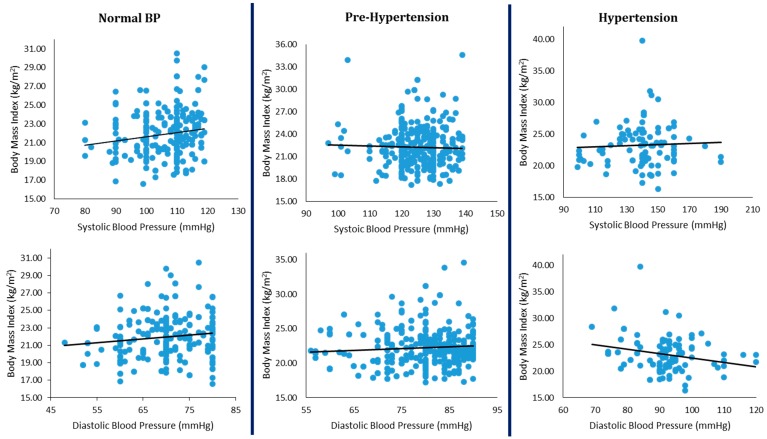
Relationships between measures of blood pressure and body mass index in the studied population.

**Table 1 medsci-05-00009-t001:** Selected anthropometric parameters of the subjects stratified by blood pressure phenotypes.

		Age (Years)	Weight (kg)	Height (m)	WC (cm)	HC (cm)	BMI (kg/m^2^)	WHpR	WHtR
**Males** (*N* = 742)	Normal BP (*N* = 262)	23.4 ± 3.6	65.7 ± 7.8	1.73 ± 0.07	78.0 ± 5.9	91.5 ± 8.4	21.9 ± 2.4	0.88 ± 0.48	0.45 ± 0.04
	Prehypertension (*N* = 367)	23.1 ± 4.1	67.8 ± 9.0	1.75 ± 0.08	79.7 ± 7.1	93.1 ± 7.8	22.2 ± 2.7	0.86 ± 0.06	0.46 ± 0.04
	*p*	0.681	0.003	0.006	0.004	0.019	0.179	0.331	0.106
	Hypertension (*N* = 113)	24.0 ± 5.8	72.0 ± 10.6	1.76 ± 0.07	83.9 ± 9.7	95.8 ± 10.5	23.3 ± 3.3	0.88 ± 0.07	0.48 ± 0.06
	*p*	0.465	<0.001	0.128	<0.001	0.003	<0.001	0.507	<0.001
**Females** (*N* = 868)	Normal BP (*N* = 454)	22.2 ± 4.7	61.9 ± 9.6	1.75 ± 0.08	79.1 ± 7.8	94.2 ± 7.8	22.8 ± 3.8	0.84 ± 0.08	0.48 ± 0.05
	Prehypertension (*N* = 313)	21.7 ± 3.8	64.0 ± 10.7	1.66 ± 0.07	81.4 ± 8.0	96.5 ± 9.3	23.3 ± 4.2	0.85 ± 0.07	0.49 ± 0.05
	*p*	0.537	0.007	0.178	<0.001	<0.001	0.072	0.526	0.009
	Hypertension (*N* = 101)	22.5 ± 4.5	72.4 ± 15.0	1.66 ± 0.08	86.8 ± 11.7	98.5 ± 10.5	26.2 ± 5.4	0.88 ± 0.10	0.52 ± 0.07
	*p*	0.533	<0.001	0.727	<0.001	0.046	<0.001	<0.001	<0.001

*p*-values below the mean ± standard deviation (S.D.) for prehypertension are for comparisons with subjects with normal BP, while those below values for hypertension are for comparisons with subjects with prehypertension. *p*-values for comparisons between values for subjects with normal BP and those with hypertension are not shown. They are, however, all <0.05 except for age in both sexes, height in females, and WHpR in males. BMI, BP, WC, HC, BMI, WHpR, and WHtR represent body mass index, blood pressure, waist circumference, hip circumference, body mass index, waist-to-hip ratio, and waist-to-height ratio, respectively.

**Table 2 medsci-05-00009-t002:** Correlations between anthropometric parameters with systolic and diastolic blood pressures stratified by blood pressure phenotypes.

			Males			Females		
			WC	BMI	WHtR	WC	BMI	WHtR
**All Subjects**	Systolic BP	*r* (*p*)	+0.229 (<0.001)	+0.138 (<0.001)	+0.162 (<0.001)	+0.232 (<0.001)	+0.169 (<0.001)	+0.189 (<0.001)
	Diastolic BP	*r* (*p*)	+0.137 (<0.001)	+0.110 (0.003)	+0.118 (0.001)	+0.230 (<0.001)	+0.261 (<0.001)	+0.246 (<0.001)
**Normal BP**	Systolic BP	*r* (*p*)	+0.189 (0.002)	+0.164 (0.008)	+0.122 (0.049)	+0.128 (0.006)	+0.050 (0.291)	+0.097 (0.038)
	Diastolic BP	*r* (*p*)	−0.007 (0.912)	+0.126 (0.041)	−0.032 (0.606)	−0.030 (0.522)	+0.045 (0.342)	−0.007 (0.879)
**Prehypertension**	Systolic BP	*r* (*p*)	+0.126 (0.015)	−0.032 (0.538)	+0.094 (0.073)	+0.152 (0.007)	+0.015 (0.789)	+0.076 (0.178)
	Diastolic BP	*r* (*p*)	+0.038 (0.472)	+0.079 (0.131)	−0.186 (<0.001)	+0.167 (0.003)	+0.242 (<0.001)	+0.234 (<0.001)
**Hypertension**	Systolic BP	*r* (*p*)	−0.055 (0.566)	+0.051 (0.593)	−0.065 (0.497)	−0.114 (0.256)	−0.033 (0.745)	−0.068 (0.500)
	Diastolic BP	*r* (*p*)	−0.209 (0.026)	−0.252 (0.007)	−0.162 (0.086)	+0.175 (0.081)	+0.255 (0.010)	+0.268 (0.007)

**Table 3 medsci-05-00009-t003:** Correlations between anthropometric parameters with systolic and diastolic blood pressures stratified by measures of obesity.

			Males		Females	
			BMI	WHtR	BMI	WHtR
**All Subjects**	Systolic BP	*r* (*p*)	+0.138 (<0.001)	+0.162 (<0.001)	+0.169 (<0.001)	+0.189 (<0.001)
	Diastolic BP	*r* (*p*)	+0.110 (0.003)	+0.118 (0.001)	+0.261 (<0.001)	+0.246 (<0.001)
**Normal Weight**	Systolic BP	*r* (*p*)	+0.077 (0.051)	+0.062 (0.115)	+0.009 (0.821)	+0.089 (0.030)
	Diastolic BP	*r* (*p*)	+0.081 (0.039)	−0.006 (0.879)	+0.074 (0.063)	+0.011 (0.788)
**Overweight**/**Obese**	Systolic BP	*r* (*p*)	+0.131 (0.212)	+0.325 (0.001)	+0.268 (<0.001)	+0.243 (<0.001)
	Diastolic BP	*r* (*p*)	+0.060 (0.567)	+0.147 (0.156)	+0.219 (0.001)	+0.223 (<0.001)

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
