# Peer review of "Not a ‘Straitjacket Affair’: Anthropometrically Derived Obesity Index Correlates of Elevated Blood Pressure among University Undergraduates"

_medsci, 2017, doi:10.3390/medsci5020009_

Reviewer 1 Report

Not a “straitjacket affair”: Anthropometrically-derived 1 obesity index-correlates of elevated blood pressure among 2 University undergraduates 3

Chukwunonso ECC Ejike1,2* and Patricia O Ukegbu3

General comments

Overweight/obesity (O/O) prevalence is increasing worldwide. First this wave occurred in developed countries and from the last two or three decades it is affecting developing countries. O/O, however, is not uniformly widespread inside countries. In general, population in the lowest socieconomic strata are more affected than the ones in the upper strata.

The relationship between O/O and blood pressure has been studied is known for a long time. In general, it is accepted that fat tissue accumulation predisposes for blood pressure increase and to hyperension development. This relationship, however, is not linear. Higher degrees of obesity does not correlate (BMI higher than 32 or 35) with blood pressure levels.

In this article authors studied the relationship between several O/O indexes and blood pressure in a large cohort of young Nigerians (18-30 y). They observed that linear correlation was found only for those with blood pressure under hypertensive levels (<140/90 mmHg). Moreover, unlike expectancy, correlations were higher for systolic and not for diastolic blood pressure. 

Specific comments

A sample size of 330 was calculated. However, more that 1,600 subjects were recruited and studied. Considering that this study was not designed to investigate prevalence but correlation between continuous variables (BMI vs BP), I suggest to remove the formula and to indicate the power of study and to clarify how individuals were selected. Authors state that a 'a randon sample' of students were selected. In this case it is necessary to add other information about recruitment (how many were selected, how many attended, etc). 

The proportion of hypertensive subjects is quite elevated considering the age range of the group. Use of antihypetensive were investigated? If so, I suggest to remove those under such drugs from analysis, because antihypertensives mask the real blood pressure value. 

I suggest to the authors to make clear all along the text that the predicted variables are the anthropometric indexes and the outcome variable was blood pressure. Sometimes authors suggest an inverse relationship. Even considering the low degree of association I suggest to include a figure containing a scatter plot of BMI vs Systolic BP (X vs Y plot) and BMI vs Diastolic BP with the best correlation. I suggest to test the linear or a polynomial model (second order). 

There are few problems with english language. I am not not native english. I suggest a revision because publication.

Author Response

Specific comments

1. A sample size of 330 was calculated. However, more than 1600 subjects were recruited and studied. Considering that this study was not designed to investigate prevalence but correlation between continuous variables (BMI vs BP), I suggest to remove the formula and to indicate the power of study and to clarify how individuals were selected. Authors state that ‘a random sample’ of students were selected. In this case it is necessary to add other information about recruitment (how many were selected, how many attended, etc).

We accept the suggestions and have modified the manuscript accordingly.

2. The proportion of hypertensive subjects is quite elevated considering the age range of the group. Use of antihypertensive were investigated? If so, I suggest to remove those under such drugs from analysis, because antihypertensives mask the real blood pressure value.

The subjects were naïve to BP measurements and none of them was taking any drugs for any ailment. The high prevalence of the population may be due to white-collar hypertension or to the use of an oscillometric device. These were discussed in the discussion section.

3. I suggest the authors to make clear all along the text that the predicted variables are anthropometric indexes and the outcome variable was blood pressure. Sometimes authors suggest an inverse relationship. Even considering the low degree of association I suggest to include a figure containing a scatter plot of BMI vs Systolic BP (X vs Y plot) and BMI vs Diastolic BP with the best correlation. I suggest to test the linear or a polynomial model (second order).

We used a dual approach. In the first instance, we used blood pressures as the outcome variables, and in the second instance we used obesity as outcome variables. The idea was simply to achieve a holistic view of the data. We would have loved to include scatter plots but there would be too many plots, leading to “data noise”. We have therefore preferred to present the correlation coefficients in tables.

4. There are a few problems with English language. I am not native English. I suggest revision because publication.

We have revised the manuscript for English language correctness.

We thank the reviewer for the kind and very useful comments. The have improved the manuscript considerably.

Reviewer 2 Report

The aim of the study was to test the hypothesis that the “plateau effect” of high blood pressure on the blood pressure versus body mass index curves of adults exists independent of the measure of obesity in question. The topic on which this manuscript builds is fairly novel and interesting, however I would suggest that the findings be presented more accurately and appropriately, as listed below 

Overall: I think the entire manuscript needs to undergo a thorough language check. 

Introduction: Page 2, Reference 4-6 related to the “plateau/hyperbole” phenomenon are only of the papers of the main author. Why is this? Are there no other studies that has been done on this topic? Furthermore, were the associations in references 4-6 age-adjusted? 

Page 2, according to the methods section the sample was chosen at random, “…yet sampling was purposive…”. How is this possible, as purposive sampling in itself is subjective sampling and thus not completely at random. 

Page 3, please define WHpR and WHtR when first mentioned 

Page 3, statistics: The statistical methodology is not clear. Firstly, according to the manuscript, all descriptive data are presented as mean±SD, which is correct if the data is normally distributed (or Gaussian). Was all the measured parameters normally distributed? If data is not normally distributed presenting the data as median (interquartile range) would be more informative. Furthermore, according to the statistical analysis section, oneway ANOVA was done to “…determine if differences between continuous data for male and female respondents were significant.” Which continuous data is referred to? And why use oneway ANOVA to determine differences in, for ex SBP, between male and female? A student t-test (parametric test) or Mann–Whitney U test (nonparametric test) would be the appropriate tests. With that said, nowhere in the manuscript are males and females compared. The use of the oneway ANOVA would be relevant to compare the differences in the continuous variables in the different BP groups (Table 1 - normotensive, pre-hypertensive and hypertensive), however a post-hoc test is needed to determine which groups are significantly different. This too has not been noted in the statistical analysis section. Was a post-hoc test done? If so, which test was done? 

Lastly, Pearson correlations were conducted however it is not specified whether these variables were normally distributed or not. Spearman correlations needs to be done on non-normal data. 

Pages 3-4, lines 122-127, would be better suited in a table (in my opinion – Table 1 Page 4, lines 131-136, the conclusions drawn from the data in this section are more suited in the discussion and not in the results section. The actual data, I think, is more suited in a table (in my opinion – Table 1), thus also remove figure 2. Page 5, lines 139-140, the p-value (significance) for this observation is not noted in the table. 

According to the footer of Table 1, the p- values below the mean ± S.D. for prehypertension are for comparisons with subjects with normal BP, while those below values for hypertension are for comparisons with subjects with prehypertension. Thus hypertensive vs. normotensive are not compared Page 5, lines 146-147, according to the text, reported anthropometric indices clearly increased significantly with the progression of BP phenotypes in the subjects, especially in the males. However, there are no p-values to substantiate this observation. 

Discussion: The discussion section does not fully explain the findings of the study. The paragraphs are too long. Paragraph 1 can be reduced substantially Lines 208-210, related to the white-coat effect. Would the white-coat effect not then be as a plausible explanation for the prevalence of hypertension? Why would only the pre-hypertensive participant be effected by the white-coat effect?

Lines 258-260, the plateau/ hyperbole relationship is not very well explained. Figures and Tables: Figure 1: remove and only put data in text. Figure 2: remove and put data in Table 1. Table 1: expand this table. Perhaps change it to the “Participant characteristics” per BP group. Why is there no comparisons done between normotensives and hypertensives? Furthermore, include age (mean or median depending on the data distribution), sample size in each group, number of males and females, anthropometric data, BP data etc.

Author Response

1. I think the entire manuscript needs to undergo a thorough language check.

We have cross-checked the entire manuscript and corrected the language problems in it.

2. Page 2, Reference 4‐6 related to the “plateau/hyperbole” phenomenon are only of the papers of the main author. Why is this? Are there no other studies that has been done on this topic?

Though there are other studies that have studied the relationship between blood pressure and overweight/obesity, only the cited papers have used the terms “plateau” and “hyperbole” in describing their relationship. We cited other papers on the subject where we felt they were more appropriate.

3. Page 2, according to the methods section the sample was chosen at random, “…yet sampling was purposive…”. How is this possible, as purposive sampling in itself is subjective sampling and thus not completely at random.

We have edited the manuscript to remove “random” as it appears “purposive” describes the method better. We in fact chose the students at random, but ensured that students from different faculties in each University were represented.

4. Page 3, please define WHpR and WHtR when first mentioned

We accept the correction and have edited the manuscript accordingly.

5a. Page 3, statistics: The statistical methodology is not clear. Firstly, according to the manuscript, all descriptive data are presented as mean±SD, which is correct if the data is normally distributed (or Gaussian). Was all the measured parameters normally distributed? If data is not normally distributed presenting the data as median (interquartile range) would be more informative.

The data were normally distributed.

5b. Furthermore, according to the statistical analysis section, oneway ANOVA was done to “…determine if differences between continuous data for male and female respondents were significant.” Which continuous data is referred to?

The continuous data are data for age, anthropometric measurements and blood pressures.

5c. And why use oneway ANOVA to determine differences in, for ex SBP, between male and female? A student t‐test (parametric test) or Mann–Whitney U test (nonparametric test) would be the appropriate tests. With that said, nowhere in the manuscript are males and females compared. The use of the oneway ANOVA would be relevant to compare the differences in the continuous variables in the different BP groups (Table 1 ‐ normotensive, pre‐hypertensive and hypertensive), however a post‐hoc test is needed to determine which groups are significantly different. This too has not been noted in the statistical analysis section. Was a post‐hoc test done? If so, which test was done?

One way ANOVA was chosen because it is appropriate for continuous variables that are normally distributed, and it allows for multiple comparisons. We have edited the data analysis section to indicate that post hoc multiple comparisons were done.

5d. Lastly, Pearson correlations were conducted however it is not specified whether these variables were normally distributed or not. Spearman correlations needs to be done on non‐normal data.

As indicated above, the data were normally distributed.

6. Pages 3‐4, lines 122‐127, would be better suited in a table (in my opinion – Table 1)

We accept the suggestions and have made the required modifications.

7. Page 4, lines 131‐136, the conclusions drawn from the data in this section are more suited in the discussion and not in the results section. The actual data, I think, is more suited in a table (in my opinion – Table 1), thus also remove figure 2.

We have removed the conclusions as we agree they are better suited in the discussion. We however prefer to let the Figure stand as the data in it (unlike that of Figure 1) cannot be inserted into Table 1.

8. Page 5, lines 139‐140, the p‐value (significance) for this observation is not noted in the table. According to the footer of Table 1, the p‐ values below the mean ± S.D. for prehypertension are for comparisons with subjects with normal BP, while those below values for hypertension are for comparisons with subjects with prehypertension. Thus hypertensive vs. normotensive are not compared.

The reviewer is right here. Though we compared hypertensive vs normotensive subjects, we did not show the p values in the Table. A close look at the values presented in the Table will show that the values for hypertensives were larger than those for prehypertensives which in turn were larger than those for normotensives. We therefore reasoned that presenting the p values for the comparisons between values for hypertensives vs prehypertensives, the reader will understand that values for normotensives, being lower than those for prehypertensives would also be significant (especially as we have made that clear in the text). We also desired to avoid “noise” in the Table by having too many values for different comparisons. Nonetheless, we have modified the footnote to the Table to clarify this point.

9. Page 5, lines 146‐147, according to the text, reported anthropometric indices clearly increased significantly with the progression of BP phenotypes in the subjects, especially in the males. However, there are no p‐values to substantiate this observation.

The significant increase in the anthropometric variables with progression from normotension through prehypertension to hypertension is evident in Table 1. The necessary p values are presented in the said Table, and clarifications made in the footnote.

10. The discussion section does not fully explain the findings of the study.

We think we have done justice to discussing our key findings.

11. The paragraphs are too long. Paragraph 1 can be reduced substantially

We have edited the manuscript and removed extraneous materials.

12. Lines 208‐210, related to the white‐coat effect. Would the white‐coat effect not then be as a plausible explanation for the prevalence of hypertension? Why would only the pre‐hypertensive participant be effected by the white‐coat effect?

We are sorry about the mistake and have edited the manuscript to reflect that white-coat hypertension may have affected both prehypertension and hypertension.

13. Lines 258‐260, the plateau/ hyperbole relationship is not very well explained.

The plateau/hyperbole is understood in the context of the entire manuscript. It describes the rise in blood pressure with obesity among normotensive which tappers in prehypertensives and flattens out in hypertensives.

14. Figure 1: remove and only put data in text. Figure 2: remove and put data in Table 1.

As mentioned earlier, we have removed Figure 1 but prefer to retain the Figure 2 as we do not wish to make Table 1 unwieldy.

15a. Table 1: expand this table. Perhaps change it to the “Participant characteristics” per BP group. Why is there no comparisons done between normotensives and hypertensives?

We have responded to these earlier.

15b. Furthermore, include age (mean of median depending on the data distribution), sample size in each group, number of males and females, anthropometric data, BP data etc.

We have modified the Table to include the data requested.

We are grateful to the reviewer for the very useful comments. They have improved our manuscript tremendously.

Round  2

Reviewer 1 Report

No more comments.

Author Response

Thank you for your very useful views and comments in the last revision cycle. We appreciate them.

Reviewer 2 Report

Thank you for clarifying my concerns. 

Author Response

Thank you for your kind comments. We have added the scatter plots, for BMI vs SBP/DBP, as requested. We are grateful to you for your prompt attention to our manuscript.